# Cannabis Pharmacogenomics: A Path to Personalized Medicine

Mariana Babayeva [1,*] and Zvi G. Loewy [1,2]

[1] Department of Biomedical and Pharmaceutical Sciences, Touro College of Pharmacy, New York, NY 10027, USA

[2] Department of Pathology, Microbiology and Immunology, New York Medical College, Valhalla, NY 10595, USA

\* Correspondence: mariana.babayeva@touro.edu

**Abstract:** Cannabis and related compounds have created significant research interest as a promising therapy in many disorders. However, the individual therapeutic effects of cannabinoids and the incidence of side effects are still difficult to determine. Pharmacogenomics may provide the answers to many questions and concerns regarding the cannabis/cannabinoid treatment and help us to understand the variability in individual responses and associated risks. Pharmacogenomics research has made meaningful progress in identifying genetic variations that play a critical role in interpatient variability in response to cannabis. This review classifies the current knowledge of pharmacogenomics associated with medical marijuana and related compounds and can assist in improving the outcomes of cannabinoid therapy and to minimize the adverse effects of cannabis use. Specific examples of pharmacogenomics informing pharmacotherapy as a path to personalized medicine are discussed.

**Keywords:** pharmacogenomics; cannabis; cannabinoids; personalized medicine





## 1. Introduction

The initiation of personalized medicine has come with the potential for improving the efficacy and safety of medications. Variations in the genes in any of the involved pathways might impact a patient's prognosis, pharmacological response, and adverse effects of therapy. Knowledge of the pharmacogenomics (PGx) of cannabinoids is necessary for effective and safe dosing and to avoid treatment failure and severe complications.

Cannabis is regulated as a schedule 1 substance by the U.S. federal government. However, 37 states, the District of Columbia, Guam, Puerto Rico, and the U.S. Virgin Islands have comprehensive medical marijuana programs with indications for a range of chronic illnesses. In addition, the remaining 13 states allow the use of cannabidiol (CBD) for medical reasons in limited situations [1].

The medicinal use of cannabis in ancient China dates to about 2700 BC [2,3]. Cannabis has a wide range of clinical applications and the list of diseases in which cannabis/cannabinoids are used as a treatment is constantly increasing. Studies in experimental models and humans have suggested anti-inflammatory, neuroprotective, anxiolytic, and antipsychotic properties of chemicals extracted from cannabis [4]. Cannabis contains more than 100 cannabinoids, where CBD and THC are the subjects of most studies [5,6]. THC is the main psychoactive constituent and can produce neuroprotective, analgesic, antiemetic, and antiglaucoma effects [7,8]. CBD decreases THC psychoactivity and exhibits anti-inflammatory, antioxidant, anticonvulsant, and neuroprotective effects [6,9–11]. CBD (Epidiolex) has been FDA and EMA approved for Dravet and Lennox–Gastaut syndromes [4]. Another cannabis medication, Sativex (THC:CBD, 1:1 ratio), is used to treat symptoms of multiple sclerosis [4]. Moreover, a synthetic pharmaceutical-grade THC (dronabinol and nabilone) has been FDA approved for the treatment of chemotherapy-induced nausea and vomiting in patients who failed to respond to traditional antiemetic therapy. Dronabinol has also been approved as a therapy for anorexia in patients with AIDS [12]. Other cannabinoids including cannabidivarin also

contribute to the medicinal effects of cannabis. Cannabidivarin, also known as cannabidivarol or CBDV, has recently gained significant attention. CBDV is the propyl analog of CBD and is similar to CBD structurally and functionally. CBDV is a nonpsychotropic phytocannabinoid with anti-inflammatory and anticonvulsant activities [13]. In October 2017, CBDV was given an orphan designation by the EMA for use in Rett syndrome and in February 2018 for the treatment of fragile X syndrome [14]. In 2020, the FDA also granted an orphan designation to CBDV for fragile X and Rett syndromes. Recently, a few clinical trials with CBDV were announced to assess the efficacy and safety of CBDV in the treatment of autism spectrum disorder (ASD) and Prader–Willi syndrome (PWS) [15].

Mechanisms of action of the cannabinoids involve interaction with the cannabinoid as well as non-cannabinoid system. THC has been shown to modulate many of its effects through the cannabinoid-1 (CB1), cannabinoid-2 (CB2), and G-protein coupled receptors (GPR55). THC is a partial agonist of these receptors [7,16–18]. In contrast, CBD has little affinity for CB1 and CB2 receptors but acts as an indirect antagonist of cannabinoid agonists and as an inverse agonist of the CB2 receptor [6,9,10]. CBD increases the concentrations of endocannabinoid anandamide (AEA) through the inhibition of its metabolizing enzyme fatty acid amide hydrolase (FAAH). AEA is an agonist at CB1 and CB2 [19]. Therefore, CBD is indirectly involved in the regulation of the CB1 and CB2 receptors. CBD may also modulate non-endocannabinoid systems including GPR55, and transient receptor potential cation channel subfamilies V, A, and M (TRPV, TRPA, TRPM) [20]. CBD acts as an agonist at the TRPV1, TRPV2, TRPV3, and TRPA1 receptors as well as an antagonist at GPR55 and TRPM8 [16,18,21–25]. Although the mechanism of action of CBDV is still unclear, it has been suggested that CBDV may produce its effects through the TRPV1 and TRPV2 receptors [26,27]. In addition, CBDV displays activity at CB2 but not at CB1 receptors [28,29].

All three phytocannabinoids (THC, CBD, CBDV) are highly lipophilic compounds, which accumulate extensively in the adipose tissues [30–32]. The absorption of THC depends on the route of administration. The lowest THC bioavailability is oral (6%). Smoked and inhaled bioavailability is 25% and 10–35%, respectively [33,34]. The differences are mostly due to the presystemic metabolism of THC in the gut wall and in the liver. THC is highly protein bound (95–99%) with a half-life of 25–36 h [30,35]. THC undergoes phase 1 hepatic metabolism by the CYP2C9, CYP2C19, and CYP3A4 enzymes to psychoactive metabolite 11-OH-THC, which further oxidizes to inactive 11-COOH-THC [36]. Even though more than 30 THC metabolites were detected, these two metabolites dominated. Both major metabolites undergo phase 2 biotransformation. The 11-COOH-THC is metabolized mostly by the UGT1A3 enzyme and 11-OH-THC is metabolized by the UGT1A9 and UGT1A10 enzymes. Most of the THC excreted in the feces (65%) and in the urine (up to 25%) is in the form of the parent compound, 7-OH-THC, THC-COOH, and various glucuronide conjugates [37].

CBD and CBDV have poor oral bioavailability (6%), similar to THC [38]. Such low bioavailability can be explained by significant first-pass metabolism and erratic absorption [39]. In contrast, intranasal CBD has a bioavailability of 34–46% [40]. The half-life of both compounds is similar at 18–32 h [37,41]. CBD is highly bound to plasma proteins (95%) and is mainly metabolized to its active metabolites 7-OH-CBD and 7-COOH-CBD by the CYP2C19, CYP2C9, and CYP3A4 enzymes [42,43]. These metabolites are then further converted into glucuronide conjugates by UGT1A9 and UGT2B7 [44]. A large portion of CBD and its metabolites are excreted in the feces (82%) and small portions are eliminated via the urine [43]. The CBDV pharmacokinetic data are insufficient. CBDV rapidly penetrates the blood–brain barrier and the plasma concentrations are lower in the plasma than in the brain [41]. CBDV is rapidly metabolized in the liver to 7-OH-CBDV and 7-COOH-CBDV, although the exact metabolic pathway is still unknown [45].

In addition, CBD and THC are substrates and inhibitors for active transport. Membrane proteins, P-gp and BCRP, interact with both cannabinoids [46].

Polymorphisms in the genes of the corresponding receptors, the enzymes, and the transporters can affect the pharmacokinetics, response, and resistance to cannabinoid therapy as well as the development of cannabis use disorders and cannabis-induced changes in executive functions.

Treatment by cannabis and cannabinoids is a part of innovative medicine. The list of medical disorders in which cannabinoids are used as a therapy is rapidly growing. However, knowledge of the medicinal effects as well as the incidences and severity of the side/adverse effects of cannabinoids is still lacking. Pharmacogenomics can help predict both positive and negative effects of cannabinoids and precisely identify the best treatment and dose for each individual, thereby reducing the complications, hospitalizations, and treatment cost. More recently, the importance of characterizing synonymous single-nucleotide variants (sSNVs) with respect to their role in regulatory functions exhibited in health and disease has gained focus [47]. A valuable resource that can be accessed for information relating human genetic variation and response to medications is the PharmGKB database [48].

## 2. Pharmacogenomics of Receptors

**CNR1**. The CB1 receptor, encoded by the *CNR1* gene, is expressed in the central and peripheral nervous systems, mainly in the cerebellum, hippocampus, basal ganglia, frontal cortex, amygdala, hypothalamus, and midbrain [49]. CNR1 is the main molecular target for THC. Activation of this receptor stimulates the appetite and has antiemetic, analgesic, and sedative effects [50]. Some genetic studies have linked polymorphisms in the CNR1 gene with an increased risk of schizophrenia [51,52]. A decrease in both *CNR1* mRNA and the receptor levels has been reported in patients with schizophrenia [53]. However, other studies have not supported this association [54]. Upregulated expression of the CNR1 gene was observed after THC exposure in patients with mood disorders [54].

Almost all genetic studies with *CNR1* were conducted to discover a link between *CNR1* polymorphism and cannabis use disorder. While some studies have not found an association between polymorphism in *CNR1* and cannabis dependance [55,56], most of the genetic studies have associated variations in the CNR1 gene with cannabis addiction [46,57–59]. Connections between the polymorphism of CNR1 and substance abuse have been reported including cannabis, alcohol, and cocaine [60–62].

SNP ID at the *rs806368, C* allele has been associated with an increased risk of cannabis dependence and with a lower expression of CNR1 in the brain [54,63]. Individuals with one or both copies of the *rs806368 C* allele had a 5.4-fold increase in the probability of frequent and persistent cannabis use [64]. Interestingly, there were substantial differences between European Americans (20%) and African-Americans (8%) in the minor allele frequencies of the genetic variation [63]. In addition, *rs806368* was found to influence substance dependence by an interaction with *rs6454674* [62]. However, a recent study reported no association of cannabis addiction and *rs806368* [65]. Limitations of this study were small size (49 cannabis addicted individuals) and the fact that all subjects belonged to the Pakistani population. No data were reported on the frequency of the variant in this population.

Another SNP, *rs806380*, was associated with the development of cannabis dependence in adolescents. Significant differences have been reported in the allele frequency between Caucasians and Hispanics. Caucasians demonstrated a significant association between *rs806380* and cannabis addiction [58,63,66]. Moreover, it was reported that the *A* allele of *rs806380* was more common in cannabis-dependent individuals, while the *G* allele (21% of the subjects) was more common in those with no cannabis obsession [58,63].

Some other *CNR1* haplotypes (*rs6454674, rs806377, rs1049353*) were associated with cannabis dependence [63]. It has been reported that *C* carriers at *rs806374* may frequently use cannabis [67]. CNR1 *rs1406977 G* carriers had reduced *CNR1* prefrontal mRNA levels and reduced working memory compared with *AA* subjects [68]. Results of the studies with SNP *rs1049353* are controversial. Some studies did not find a significant association of *rs1049353* with abused substances or cannabis dependence [58,62,69]. However, one

study demonstrated a significant alliance of *1359AA* with protection from heroin addiction in Caucasians [70]. In contrast, a significant association of the homozygous *AA* genotype with severe alcohol dependence in the Caucasian population has been reported [71]. The *1359 G/A* of the *CNR1* gene is a common polymorphism in Caucasian populations. It was reported that 51.9% of Caucasians had the wild genotype *G1359G* and 48.1% patients had the variant genotypes *G1359A* (39.9%) or *A1359A* (8.2%) [72]. *CNR1* mutations are uncommon in the African-American population [73]. A substantial connection has been reported between *rs1049353* of the *CNR1* gene and cannabis disorder [60,74]. The *G* allele and homozygous *GG* genotype of *rs1049353* were significantly higher among cannabis users compared to control subjects [60]. This finding is consistent with the results of other studies that found an association between the *G* allele, SNP *rs1049353*, and cannabis dependence [66,75]. *CNR1 rs1049353 GG* carriers showed increased repletion after THC and THC + CBD administration compared to the placebo [76]. In addition, the *rs1049353* and *rs2023239* minor allele carriers had enhanced subjective effects during acute cannabis intoxication [77].

Another SNP, *rs2023239*, has been associated with cannabis-related phenotypes [78]. The effect of the *rs2023239* SNP genotype was moderated by the presence of the *TT* haplotype. The *C* carriers had lower levels of cannabis-related problems compared to *TT* homozygotes [79]. However, the administration of THC produced high levels of anger-hostility in *C* carriers of *rs2023239*, suggesting that mood conditions after cannabis use depend on genetic variations [80]. In addition, *rs2023239 G* cannabis users had a lower volume of bilateral hippocampi relative to the controls [81]. It has been suggested that heavy cannabis use in connection to the *CNR1 rs2023239* variation may be responsible for a small hippocampal volume [81]. Moreover, the haplotype of CNR1 *rs806368-rs1049353-rs2023239-rs6454674* and level of cannabis exposure were associated with decreased volume of the brain right anterior cingulum [82].

**CNR2**. CB2 receptors encoded by *CNR2* are highly expressed in peripheral tissues, particularly in the immune system, and at low levels in the brain glial cells such as microglia and astrocytes, and specific subpopulations of neurons [83]. Genomic studies on *CNR2* and cannabis use disorders are limited. Polymorphisms in the *CNR2* gene have been linked to pain, autoimmune disorders, and depression in humans [84]. A significant correlation was observed between variations at *CNR2 rs2501432* and depression [85]. The *CNR2 rs3003335* and *rs6658703* were associated with psychiatric comorbidities in anorexia nervosa patients. Carriers of *rs3003335 AA* and *rs6658703 GG* genotypes had higher scores in the positive symptom distress index (PSDI) and increased hostility in patients [86]. *CNR2 rs75459873* has been correlated with distressing psychotic experiences, but not with cannabis use [87].

The polymorphisms at positions *63* and *316* of the *CNR2* gene were associated with changes in the CNR2 function and altered interaction of the receptor and its substrates [88]. The *R63* allele of *rs2501432*, the *C* allele of *rs12744386*, and the haplotype of the *R63-C* allele were significantly increased among patients with schizophrenia [89]. One study demonstrated a link between the polymorphism *Q63R* and alcohol dependence in the Japanese population [90]. In rats, alcohol produced a significant downregulation of the striatal *CNR2* mRNA [91]. The low CB2 receptor expression was linked to an increased risk of schizophrenia [89,92]. Moreover, significantly lower CB2 receptor mRNA and protein levels were found in the human brain with the *CC* and *CT* genotypes of *rs12744386* compared with the *TT* genotype [89,93]. Interestingly, cannabinoid withdrawal produced a substantial CNR2 downregulation [94]. The administration of CBD blocked the reduction in CNR2 gene expression, suggesting that withdrawal disturbances can be improved by CBD [94,95]. Another study demonstrated a positive association between *CNR2 rs2501431* and cannabis use [60]. A statistically significant association has also been reported between SNPs *rs35761398* and *rs12744386* in the *CNR2* gene and cannabis dependence and schizophrenia in the Spanish population [96]. Recently, SNPs in the same variants (*rs12744386* and *rs35761398*) were correlated with a high risk of schizophrenia in patients with cannabis dependence [97].

**TRPVs.** Cannabinoids act on many molecular targets including TRPVs, TRPA1, TRPM8, and GPR55. However, no direct studies investigating the role of the *TRPVs*, *TRPA1*, *TRPM8,* and *GPR55* genetic variations on the effects of cannabis/cannabinoids have been conducted to date.

The TRPV channels (encoded by *TRPVs* genes) are mainly responsible for heat and pain detection [11]. *TRPV1* is expressed in sensory neurons and is important for thermal and chemical nociception [98]. Many SNPs have been identified in the human TRPV1 gene [99,100]. Variations in *TRPV1* (*R557K* and *G563S*) severely affect all aspects of channel activation and lead to spontaneous activity [101]. *TRPV1* variants were also linked to altered pain perception. Studies have been conducted to identify a connection between *TRPV1* polymorphism and sensitivity to capsaicin. Capsaicin stimulates burning pain, heat, and serves as a substitute model for pain. It was estimated that the *TRPV1 1911A>G* variant was related to significantly high capsaicin sensitivity [102,103]. In contrast, neuropathic pain patients carrying the *TRPV1 1911A<G* variants showed reduced capsaicin sensitivity [104, 105]. *TRPV1 1911A<G* was considered as a loss-of-function phenotype [100,106] while *TRPV1 1103C>G* was recommended as a gain-of-function phenotype [107].

Some of the *TRV1* variants were associated with differences in the disease-related properties [102,108,109]. The SNP of TRPV1 rs222741 was correlated with migraines in the Spanish population [110]. Another polymorphism of *TRPV1*, *rs8065080*, was connected to a risk of hypertension [111]. The variation of *TRPV1*, *rs4790522*, was associated with a higher salt recognition threshold in people with hypertension and obesity [111]. A significant association was reported for *TRPV1* SNP *rs222747* and tumor necrosis factor (TNF) levels in the cerebrospinal fluid of MS patients. The *TRPV1* SNP *rs222747* was connected to reduced levels of TNF [112]. Lowered TNF concentrations were associated with improved symptoms of encephalomyelitis [113,114]. *rs222747* also influences protein receptor expression and function, cortical excitability in healthy humans, and modulates pain in MS patients [107,112,115,116]. *TRPV1* together with *TRPA1* modulate airway inflammation and cough [117,118]. Polymorphisms in these genes have been correlated with childhood asthma and chronic cough [109,119,120].

Studies have demonstrated a link between *TRPV* gene polymorphisms and fibromyalgia (FM). It was reported that certain *TRPV2* haplotypes may have a protective role against fibromyalgia and some genotypes of *TRPV3* contribute toward the symptoms of FM [110]. Patients with the *AA* genotype of TRPV2 *rs1129235* were more likely to have this disease [110]. Another variant of *TRPV2 rs14039 GG* significantly increased the risks of the development of type 2 diabetes mellitus and Hashimoto thyroiditis disorders. However, the *rs4792742* variant had a strong protective effect against both conditions [121].

Polymorphisms in the *TRPV3* gene are associated with various skin diseases including atopic dermatitis and rosacea [122–124]. The variations in *TRPV3* may also have relevance to scleodactyly and tapered fingers [123,125]. Upregulated *TRPV3* activity leads to severe keratoderma and an intolerant itching sensation [126,127]. A homozygous gain-of-function *1562G>C* variant of the TRPV3 may be involved in the development of Olmsted syndrome [123]. Individuals with Olmsted syndrome also have *the following mutation variants:* TRPV3 *Gly573Ser* and *Trp692Gly* [126]. The *Trpv3 G573S* was correlated with hair loss and reduced sensitivity to cold and sharp mechanical pain [124,127].

**TRPM8.** The TRPM8 is mostly expressed in prostate tissue and dorsal root ganglia and trigeminal ganglia. The TRPM8 receptor is the primary cold receptor of the peripheral nervous system [128]. A significant association was found between cold pain feeling and the *rs12992084* polymorphism of the *TRPM8* gene [129].

Expressions of *TRPM8* mRNA and proteins are upregulated in the respiratory tract of asthma and COPD patients [130,131]. The *GC* genotype and *C* allele of *TRPM8 rs11562975* were associated with cold-induced airway hyperresponsiveness, severe bronchial obstruction, and a decline in lung function in asthmatic patients [132–134]. Other polymorphisms at *rs2052030* significantly affect susceptibility to COPD and pulmonary hypertension [135–137].

Moreover, the *rs12472151*, *rs11562975*, and *rs28901637* polymorphisms of the TRPM8 gene were associated with metabolic syndrome, obesity, and cholesterol levels [138–140]. Variants at TRPM8, *rs10166942*, and *rs2362290* have been related to slower colonic transit rates, increased risk of **irritable bowel syndrome, and chronic migraine** [141,142].

**TRPA1**. TRPA1 is a calcium-permeable cation channel expressed in sensory neurons, endothelial, and inflammatory cells [143]. *TRPA1* is upregulated in response to inflammation and chronic pain [144,145]. Some SNPs increase chemical sensitivity and channel activity of this receptor [101,146]. The gain-of-function *TRPA1* variants *797T*, *Y69C*, *R852E*, and *N855S* have greater sensitivity to agonists and an increased receptor activity than the more common allele *797R* [147–150]. However, variants *E854R* and *K868E* of *TRPA1* demonstrated dramatically reduced activity [149,150].

*TRPA1* plays a vital role in reactive airway diseases [151,152]. ALSPAC (Avon Longitudinal Study of Parents and Children) has provided strong evidence for an association between six SNPs at the *TRPA1* gene and asthma (*rs959974*, *rs1384001*, *rs7010969*, *rs3735945*, *rs920829*, and *rs4738202*) [151]. The *TRPA1* polymorphisms also contribute to variations in the control of asthma symptoms including airway inflammation and cough [120,152,153]. The *TT* genotype of the *TRPA1 rs7819749* was significantly associated with a higher degree of bronchial obstruction [133]. A significant correlation was found between *CpG-628* and *CpG-412* of *TRPA1* and pain levels [154–156]. The *TRPA1 rs920829* and *CGAGG* haplotypes were related to acute pain crisis and utilization rate (number of emergency department/acute care center admissions) in sickle cell disease patients [157]. In Spanish patients with neuropathic pain, the *G* allele and *GG* genotype in the *rs11988795* variant were protective against pain, while the *TT* genotype in the *rs13255063* variant could be a risk factor for the neuropathic pain [158]. Additionally, polymorphisms in the *TRPA1* gene were associated with paradoxical heat sensations in neuropathic pain patients [159–161]. Other polymorphisms at *TRPA1* were related to migraine and chronic fatigue syndrome (*rs2383844* and *rs4738202*) [162,163].

**GPR55.** GPR55 is a G-protein-coupled receptor that has been identified as a new cannabinoid receptor. GPR55 has little amino acid identity to the cannabinoid CB1 and CB2 receptors [164]. Given the wide localization of GPR55 in the brain and the peripheral tissues, this receptor controls multiple biological actions [165]. GPR55 interacts with exo- and endogenous cannabinoids [17,63]. Data on the interaction of the GPR55 polymorphism and cannabis/cannabinoids are limited. Based on gene association studies, the GPR55 gene has an influence on cannabis use disorder [63,166].

A reduce-of-function *584G>T* polymorphism of *GPR55* was associated with an increased incidence of anorexia nervosa in Japanese women [167]. This mutation decreased but did not eliminate GPR55 activity [168]. A recent study connected *GPR55* polymorphisms with osteoclast formation. Moreover, treatment with CBD significantly reduced bone resorption, indicating the effect of cannabinoids on osteoclasts and bone turnover [169].

*GPR55* polymorphisms have been associated with different types of cancer [168,170–172]. The overexpression of *GPR55* promoted cancer cell proliferation [164]. Upregulation of *GPR55* mRNA expression was also reported in intestinal inflammation [173]. The overexpression was associated with the development of Crohn's disease [164,174,175]. The upregulation of *GPR55* expression may also play a role in obesity [176]. The highest *GPR55* expression documented was in diabetic patients [168]. Additionally, a link between the upregulation of *GPR55* and mental disorders has been reported [177]. Interestingly, in genetic models of Rett syndrome, treatment with CBDV rescued behavioral and brain alterations including the brain weight and repaired the compromised general health status, the sociability, and motor coordination [177]. A recent genome-wide study found that a mental disorder borderline personality disorder (BPD) and life adverse events were associated with the methylation status of several genes including *GPR55* [178].

The assessment for SNPs and other genetic variants in receptors is of keen interest in pharmacological research because the identification and characterization of receptor variants may be the key to elucidating why a candidate drug acts in a quantitatively or

qualitatively different way in different people. More clinical validation is needed with cannabis receptor polymorphisms.

## 3. Pharmacogenomics of Metabolism

### 3.1. Phase 1 Metabolism

The therapeutic outcomes and adverse effects of cannabis-containing medications and cannabis depend on concentrations of the cannabinoids in the blood. The plasma levels of the cannabinoids are regulated by metabolizing enzymes. Interindividual differences in the expression and function of the corresponding enzymes may considerably affect the concentrations of the cannabinoids and their metabolites. Cytochrome P-450 (CYP-450) enzymes are major contributors to the phase I metabolism of cannabinoids. THC is metabolized by CYP2C9 and CYP3A4; CBD by CYP2C9, CYP2C19, and CYP3A4. The exact metabolic pathway of CBDV is still unknown.

**CYP2C9**. The CYP2C9 enzyme metabolizes up to 20% of medications [179]. Both THC and CBD are metabolized by this enzyme. The two most frequently occurring genotypes of the *CYP2C9* gene in populations of European descent are *CYP2C9*2* and *CYP2C9*3* [180,181]. Genetic studies have shown that *CYP2C9*2*, *3* genotypes have high frequencies in Caucasians (up to 18%) and low rates in African-Americans (1–2%) and most Asians, suggesting that these variations may be of little or no relevance in the latter populations [182–186]. These variations exhibit reduced enzyme activity and therefore produce poor metabolism of their substrates. In comparison to the normally functioning *CYP2C9*1* genotype, *CYP2C9*2* and *CYP2C9*3* are associated with approximately 30–40% and 80–90% less metabolizing power, respectively [187]. The metabolism of THC and CBD can be significantly reduced in carriers with *CYP2C9*2* or *CYP2C9*3* variants, especially in individuals that are hetero- or homozygous for the CYP2C9*3 genotype, or homozygous for the *CYP2C9*2* genotype. These poor metabolizer phenotypes suggest a low transformation rate of THC into active metabolite 11-OH-THC, and therefore a high THC/11-OH-THC concentration ratio. Interestingly, the THC/11-OH-THC ratio from a psychotic who was a poor *CYP2C9* metabolizer was the highest (1.6 vs. 0.3–1.3) among drivers suspected of driving under the influence of psychotropic drugs [188]. However, since both 11-OH-THC and THC are psychoactive compounds, changing their ratio should only have a limited effect on the appearance of psychotic symptoms. Individuals with *3 genotypes may have up to 300% higher THC levels and a 3-fold increased area under the curve (AUC) of THC and 70% lower concentration of inactive metabolite 11-COOH-THC in *CYP2C9*3/*3* homozygotes compared with wild *CYP2C9*1/*1* homozygotes [182]. A recent study confirmed significantly lower 11-COOH-THC concentrations for *CYP2C9*3* and a trend to lower 11-COOH-THC concentrations for *CYP2C9*2* carriers as well as significantly higher values of the ratio THC/11-COOH-THC for both carriers [189]. The data suggest that *CYP2C9* polymorphisms may affect the formation of both active (11-OH-THC) and inactive (11-COOH-THC) metabolites. High THC and low 11-COOH-THC concentrations can predispose the individuals to negative psychoactive effects [182]. *CYP2C9*3* carriers have demonstrated a trend toward increased sedation after THC administration [190,191]. Changes in the formation of active metabolites do not significantly change the negative impacts of THC, however, the effect can last for a longer time. The reduced formation of inactive metabolites makes the adverse effects of cannabis more dangerous.

Some other allelic variants CYP2C9*5, 6, *8, *9, *11, *13, *14 have been associated with reduced enzyme activity. *CYP2C9*5, *6, *8*, and *11 produce a decrease in the metabolism of warfarin. The Clinical Pharmacogenetics Implementation Consortium (CPIC) recommends a reduction in the warfarin dose by 15–30% per variant allele in the case of CYP2C9*5, *6, *8, or *11 [192]. However, the impacts of some of the allelic variants are not always obvious and may be substrate specific. For instance, *CYP2C9*8* produced a decrease in the metabolism of warfarin and phenytoin, an increase in the metabolism of tolbutamide, and had no effect on losartan biotransformation [193]. While the *CYP2C9*2* and *3* polymorphisms are less common in African descent, CYP2C9*5, *6, *8, and *11* have greater implications in this

population [187]. *CYP2C9*14* was almost uniquely identified in South Asians [183]. No data are available on the effect of these variants on the metabolism of cannabinoids.

Most of the individuals with poor metabolizer phenotypes were predisposed to the development of psychosis and memory impairment, especially with higher doses and/or longer durations of THC use [192,194].

A study demonstrated that the inhibition of CYP2C9 reduced the CBD metabolite (7-OH-CBD) formation to a greater extent than CYP2C19 inhibition in the *CYP2C19*1/*1* and *CYP2C19*2/*2* donors, suggesting a significant contribution of *CYP2C9* to CBD elimination [43]. However, no information is thus far available on the effect of *CYP2C9* SNPs on CBD pharmacokinetics.

**CYP3A4.** Another enzyme involved in the metabolism of THC and CBD is CYP3A4 [46]. *CYP3A4* controls the metabolism of more than 70% of all drugs [195,196]. The genetic impact on CYP3A4 activity accounts for 66% to 88% of the interindividual variations in the plasma levels and therapeutic response to substrates of CYP3A4 [197,198]. The first documented *CYP3A4* polymorphism was variant CYP3A4*1B. CYP3A4*1B carriers have demonstrated a higher drug clearance for anti-cancer agents compared to wild-type subjects [199,200]. This variant occurs in Caucasian populations at 2–9% frequencies, at higher rates in Africans (27%) [197] and was not detected in the Asian population [201]. Based on the available information, the alteration of the CYP3A4 metabolism due to the *1B variant is difficult to discover in an Asian population.

The *CYP3A4*2*, *CYP3A4*11*, *CYP3A4*12*, and *CYP3A4*17* are the most common polymorphic genotypes with reduced enzyme activity [11,202]. *CYP3A4*4* and CYP3A4*22 were also associated with reduced *CYP3A4* mRNA levels and decreased enzymatic activity [11,203–205]. The effect of the CYP3A4*22 variant accounted for 7% of the mRNA expression variability [197]. Studies have reported that the CYP3A4*22 allele plays an important role in the reduced metabolism of statins, tacrolimus, cyclosporine, and pazopanib [206–208]. There are contradictory data on the effect of CYP3A4*22 on the metabolism of voriconazole [209–211]. The authors explained this inconsistency as due to CYP3A4 having a limited effect on voriconazole metabolism, and that lower voriconazole concentrations were significantly associated with the *CYP2C19*2* polymorphism [210]. The occurrence of CYP3A4*22 in the global minor allele frequency was 2.1% [197]. The low occurrence restricts a wide contribution of *22 to the overall CYP3A4 variability. Another variant *CYP3A4 rs4646450* was also associated with the decreased protein expression and activity of CYP3A4, explaining about 3–5% of hepatic variability [208]. CYP3A4 *rs4646437* polymorphism was related to the risk of hypertension, HIV, and some types of cancer [205,212,213]. The CYP3A4 rs4646437 is highly prevalent among African and Asian populations, but not among Europeans [184].

Some *CYP3A4* variants were associated with drug addiction and withdrawal symptoms. A SNP CYP3A4 rs2242480 was significantly linked to drug addiction in the Chinese population [214]. Another variant *CYP3A4 rs4646440* was highly correlated with withdrawal symptoms and adverse reactions in methadone maintenance patients [215]. A recent study reported that *rs3735451*, *rs4646440*, and *rs4646437* had a significant correlation with decreased risk of drug addiction [196]. Unfortunately, no data are available on the effect of *CYP3A4* polymorphisms on the metabolism of cannabinoids.

**CYP2C19.** Genetic polymorphisms of *CYP2C19* significantly affect many drugs such as tricyclic antidepressants, selective serotonin reuptake inhibitors, voriconazole, clopidogrel, and more [216]. Among the *CYP2C19* polymorphisms, genotypes CYP2C19*2,*3, *4,*6,*10, and *CYP*2C19*17 are the common variants responsible for interindividual differences in the pharmacokinetics and response to CYP2C19 substrates [217]. The gain-of-function genotype, *CYP2C19*17*, has been associated with the increased production of the clopidogrel active metabolite, enhanced inhibition of platelet aggregation, and increased the risk of bleeding in patients [187,218,219]. The loss-of-function genotypes *CYP2C19*2* and *3 are responsible for the reduced metabolism of clopidogrel, decreased formation of active metabolite and antiplatelet activity, and an increased risk of adverse cardiovascular

events [220–222]. The impact of other loss-of-function variants *CYP2C19\*4* and *\*5* has not been clearly defined. The SNP *CYP2C19\*10* allele has significant clinical implications. The *CYP2C19\*10* allele has decreased enzymatic activity (up to 75%) compared to the wild-type [223]. Moreover, the *\*10* allele interferes with certain *CYP2C19* genotyping assays (*CYP2C19\*2* TaqMan assay), leading to misidentifying *CYP2C19\*10/\*2* as *CYP2C19\*2/\*2* [223]. This is essential, since the *\*10* variant maintains some metabolizing activity, but the *\*2* variant does not.

A recent study demonstrated that both CYP2C19 and CYP2C9 enzymes are important contributors in CBD metabolism to the active metabolite 7-OH-CBD [43]. However, 7-OH-CBD formation was not associated with the CYP2C19 genotype [43]. The polymorphism of the *CYP2C19* gene did not impact the THC plasma concentrations [189]. This can be explained by the small proportion of the CYP2C19 enzyme in the metabolism of THC. The catalytic activity of the CYP2C19 enzyme for THC hydroxylation was less than 2% [189,224].

The polymorphism frequency of *CYP2C19* depends on genetic ancestry. The *CYP2C19\*2* allele frequency is 36.8% in Indians, 28.4% in Asians, 16% in African-Americans, and 13.3% in Caucasians [225]. The distribution of CYP2C19\*3 showed greater variations in Indians (1.9%), Asians (10.1%), and Caucasians (0.2%) [184,225,226]. The *\*10* variant was less common, with frequencies of 0.8%, 0.25%, and 0% in African-Americans, Hispanics, and Caucasians, respectively [216,223,227]. The *CYP2C19\*17* allele is common in Caucasians (18%), African-Americans (18%), and Hispanics (15.2%), but not in Asians (4%) [184,227–229].

### 3.2. Phase 2 Metabolism

During phase 2 metabolism, the cannabinoids undergo UGT glucuronidation. The THC major metabolites are transformed mostly by the UGT1A3, UGT1A9, and UGT1A10 enzymes into glucuronide conjugates [37]. The CBD metabolites are converted into glucuronide conjugates by UGT1A9 and UGT2B7 [44]. However, glucuronidation activity toward CBD is limited and the UGT enzymes produce a minimal amount of a glucuronidated CBD product [230]. Consequently, genetic polymorphisms in UGT enzymes are unlikely to affect CBD metabolism to a major extent.

**UGT1A9.** The UGT1A9 enzyme catalyzes the conjugation of endogenous estrogenic and thyroid hormones, acetaminophen, SN-38 (an active metabolite of irinotecan), phenols, and some other compounds [231]. Many studies have shown the variable activity of the UGT1A9 enzyme. The alleles of *UGT1A9\*3*, *\*4*, and *\*5* have been associated with the reduction/elimination of the enzymatic activity of the UGT1A9 enzyme [46,231–233]. The *UGT1A9\*3* allele had 3.8% of the activity of the *UGT1A9\*1* allele and produced a significant decrease in the glucuronidation of irinotecan [232]. *UGT1A9\*3* is detected only in Caucasians and 4.4% of the population tested was found to be heterozygous (*\*1/\*3*) [232]. The decreases in enzyme activities by *UGT1A9\*5* were greater than for common variants of *UGT1A9*. The allele frequency of *UGT1A9\*5* is relatively rare (up to 0.009 in Japanese patients and 0.005 in Asian-Americans) [234,235].

Another variant, *UGT1A9\*1b*, leads to increased enzyme expression and glucuronidation rates in cancer patients treated with irinotecan [236]. This allele is found predominantly in the Asian population [236]. Some studies have associated UGT1A9 rs2741049 and rs6731242 SNPs with enhanced enzyme activities [237–239]. Some other SNPs have also been correlated with increased enzyme activity of UGT1A9 [240]. SNP variants were identified in 19% of the patients [240]. In other studies, a significant increase in propofol concentrations, AUC, and adverse effects were explained, at least in part, by the presence of the *UGT1A9 440C>T/331T>C* genotype [241,242]. Patients with UGT1A9 *440C/T* CC exhibited higher effect-site concentrations and positive efficacy compared to patients with UGT1A9 *440C/T* CT and TT [243].

Data on the association of *UGT1A9* polymorphism with the metabolism of cannabinoids are limited. A recent study demonstrated significantly lower 11-OH-THC concentrations of homozygote carriers of the derived alleles in *UGT1A9 440/331* compared with homozygote carriers of the ancestral alleles [244,245].

**UGT1A3**. UGT1A3 has a glucuronidation activity toward quercetin, luteolin, kaempferol, estrone, flavonoids, and other compounds [246]. The *UGT1A3* variants have demonstrated different activity, depending on the substrates. [247]. The metabolic actions of two *UGT1A3*2* and *5* alleles were remarkably lower than that of *UGT1A3*1* in the metabolism of quercetin, luteolin, kaempferol, flavonoids, and estrone [246]. However, in other studies, carriers of the *UGT1A3*5* and UGT1A3*2 allele produced a significantly lower valproic acid, montelukast, atorvastatin, and mitiglinide plasma concentrations, suggesting an increased activity of these variants [239,247–250]. *UGT1A3*3* produced a mild increase in estrone glucuronidation [246]. Another variant, *UGT1A3*4,* showed a 464% increase in the total glucuronidation efficiency in a Han Chinese population but decreased activity in flavonoids in a Japanese population [246,251]. It was reported that carriers of the *UGT1A3 CC* diplotype may have substantially increased expressions of *UGT1A3* mRNA and protein, and greater *UGT1A3* catalytic activity, compared with carriers of the *TT* diplotypes [252]. This information is useful to explain the published inconsistency in the metabolic activity of *UGT1A3* variants. The allele frequency distributions of the SNP *UGT1A3* in the Chinese population were statistically different to Caucasians [246]. *UGT1A3*2* has a lower frequency in the Chinese than Caucasian population, whereas *UGT1A3*4* is distributed more widely in the Chinese population than in Caucasians, but significantly less than in the Japanese population [246].

Another UGT1A variant, *rs28898617*, has been linked to increased bladder cancer risk. The risk-associated was related to increased UGT1A3 expression. This allele was only observed in the Asian population, but monomorphism was also observed in the Europeans. The total allele frequency was estimated to be 0.003 [253].

Data are lacking on the effect of UGT1A3 on the metabolism of THC.

**UGT1A10.** The UGT1A10 enzyme metabolizes steroids, bilirubin, hormones, mycophenolic acid, coumarins, quinolines, and some other compounds [254]. Interestingly, the *UGT1A10* gene is exclusively expressed in the intestine, with defective expression in the liver [255]. The allelic variant *UGT1A10*2* was associated with reduced metabolic activity and a risk of orolaryngeal cancer [256,257]. This polymorphism was prevalent in African-Americans (0.05) and less prevalent in other racial groups including Caucasians (0.01) and Asians (0.01) [258]. Another variant *UGT* (*1271, C>G*) was not linked to the alteration in the functional effect. However, this polymorphism could result in upregulated *UGT1A10* gene expression [256]. No studies have reported on the influence of *UGT1A10* polymorphism on the metabolism of THC.

**UGT2B7.** The *UGT2B7* enzyme glucuronidates many therapeutic drugs including opioids (e.g., codeine, morphine, naloxone), anticancer drugs (e.g., epirubicin), and non-steroidal anti-inflammatory drugs (e.g., diclofenac, naproxen) [259]. *UGT2B7*2 (rs7439366)* is the most common functional genetic variant with reduced enzyme activity [260]. Patients with *UGT2B7*2* polymorphisms had a significantly higher concentration and exaggerated efficacy of valproic acid compared to the wild-type genotypes [261,262]. This polymorphism was also associated with the altered metabolism and analgesic effects of morphine, fentanyl, and buprenorphine [263–266]. The *UGT2B7*2/*2* variant was correlated with a high toxicity of opioids [267,268]. However, it was also reported that *UGT2B7*2* had no effect on response to some drugs [269–271] or was correlated with higher activity of the enzyme [263,264,272,273]. This was explained by regioselectively changing the metabolites of the *UGT2B7*2* substrates [272,273]. The effect of UGT2B7*2 on drug metabolism, most probably, is substrate specific [260,274–277]. The *UGT2B7*2* variant allele was significantly rarer in the Chinese than in Caucasians and Africans [278]. The prevalence of *UGT2B7*2* was 21% in Africans and 28–52% in North Americans [235]. Other SNPs in the *UGT2B7* gene also contribute to the altered glucuronidation of drugs. The SNP *UGT2B7 rs7662029 AA* produced a higher concentration of buprenorphine compared to *GG* carriers. Additionally, a significant association was discovered between UGT2B7 *rs7662029* and increased the craving and withdrawal symptoms in heroin addict patients [264]. The enzyme activity of *UGT2B7-1 T/T* on mitiglinide metabolism was stronger than that of other genotypes [279]. The *UGT2B7*1a* allele was also significantly associated with altered efavirenz metabolism.

*UGT2B7*1a* produced 41% higher efavirenz concentrations [280]. Another *UGT2B7-161CC* polymorphism had lower metabolic activity and may produce more significant drug efficacy compared to other carriers [259]. Patients with *UGT2B7-211* (*GT* and *TT*) genotypes demonstrated lower substrate plasma concentrations than the wild-type [259,281]. The SNP *211G > T* was present only in Asian-Americans (9%) and Hispanic-Americans (2%) [235].

Data are lacking on the effect of *UGT2B7* polymorphisms on CBD metabolism. CBD glucuronidation has a reduced role in the overall elimination of the drug. Most probably, genetic variations at *UGT2B7* are unlikely to affect CBD metabolism to a major extent.

### 3.3. Metabolic Drug-Drug Interactions (DDI)

CBD, THC, and other cannabinoids are susceptible to metabolic drug–drug interactions, as the cannabinoids are not only substrates but also inhibitors and/or inducers of several metabolic enzymes. Medications that are prominent substrates for these enzymes may be at risk of altered elimination and pharmacologic response by concomitant use of the cannabinoids. Moreover, undesirable DDIs with xenobiotics may occur in co-users of cannabis.

CBD can be involved in strong drug interactions mediated by CYP2C9, 2C19, and 3A and moderate drug interactions mediated by CYP1A2 and 2D6. THC may participate in strong CYP2C9 and weak CYP1A2 and 3A mediated drug interactions [282–284]. For example, the oral administration of CBD with the anticonvulsant clobazam led to a significant increase in the plasma concentrations and AUC of its active metabolite N-desmethylclobazam, which is metabolized predominantly by CYP2C19 [285–287]. A case report with warfarin (mainly metabolized by CYP2C9) demonstrated that the patient's international normalized ratio (INR) was increased from 1.8 to 11.55 because of frequent cannabis smoking [288].

Moreover, CBD and THC demonstrate strong inhibition of the metabolic activities of the non-CYP enzymes UGT1A6, 1A9, 2B4, and 2B7, and insignificant inhibition of a number of additional UGTs including UGT2B17 [289]. CBD has been shown to be a more potent inhibitor compared to THC as the $IC_{50}$ values of CBD were 2–3-fold lower than that observed for THC [289]. The administration of midazolam with epidolex (CBD) resulted in increased plasma concentrations, AUC, and half-life of active midazolam metabolite 1-hydroxymidazolam [290]. Although midazolam itself is not glucuronidated by UGT2B7, its active metabolite, 1-hydroxymidazolam, is a UGT2B7 substrate [289,291].

However, DDI can also alter the pharmacokinetics of cannabinoids as well as their therapeutic/adverse effects. The PK of the oromucosal spray Sativex® (nabiximols, THC to CBD ratio is 1:1) was investigated in combination with rifampicin (CYP3A and 2C19 inducer) and ketoconazole (CYP3A inhibitor). Rifampicin reduced the Cmax and AUC of both cannabinoids. Rifampin decreased Cmax by 36%, 52%, and 87% for THC, CBD, and 11-OH-THC, respectively. In contrast, ketoconazole co-administration increased the Cmax of the THC, CBD, and 11-OH-THC by 27%, 89%, and 204%, respectively [292]. Therefore, potential effects should be taken into consideration when co-administered with THC and/or CBD containing medications with inhibitors or inducers of the cannabinoid metabolic pathways. The interactants can also exaggerate/diminish the effects of smoking cannabis.

Most of the DDI with cannabinoids are pharmacokinetic interactions, resulting in altered plasma levels of one of the interactants. However, the structure of cannabinoid–opioid interactions remains undiscovered. Studies have reported that vaporized cannabis increased the analgesic effect of morphine and oxycodone without producing significant differences in the AUC of the medications in patients with chronic pain [293,294]. It was suggested that there is a pharmacodynamic interaction between the opioids and cannabinoids, which most probably involve altered interactions with receptors [284]. All drug–drug interactions are complex; however, genetic variations can make the DDIs even more problematic and risky.

## 4. Pharmacogenomics of Transport

THC has an affinity for two membrane proteins, ABCB1(P-gp or P-glycoprotein) and ABCG2 (BCRP) [46]. CBD is not a substrate for these transporters [295]. However, both CBD and THC inhibit P-gp and BCRP proteins [296–298]. As an inhibitor of these efflux transporters, CBD might modulate the brain disposition of THC, which could explain, in part, its known ability to modulate THC psychoactive effects.

**ABCB1**. The P-gp is an efflux protein belonging to the ATP-binding cassette subfamily B member 1 (ABCB1). Substrates of the transporter are various structurally unrelated compounds such as xenobiotics, endogenous compounds, steroid hormones, lipids, phospholipids, cholesterol, cytokines, pharmaceuticals, neutraceuticals, dietary, and other compounds [299,300]. ABCB1 limits the absorption of xenobiotics, reduces their expression in tissues, and is also involved in the biliary and renal elimination of its substrates. Polymorphisms of the *ABCB1* gene are associated with alterations in the pharmacokinetics of some drugs, resistance to drug treatment, and susceptibility to numerous diseases [301].

In recent years, a few polymorphisms of the *ABCB1* gene have been described. The variants *Gly412Gly*, *rs1128503*, *rs2032582*, and *rs1045642* are the most common polymorphisms of the *ABCB1* gene. The three SNPs exhibit the highest frequencies in Asian and Caucasians populations and the lowest in African populations [299].

The results of studies investigating the effects of *1236C>T* polymorphisms at the *ABCB1* gene were inconsistent. Studies found an increased drug level and/or drug effect in both the *1236 CC* genotype and the *1236 TT* genotype [302–304] or no genetic effect at all [300,305]. The allele frequency for SNP *rs1128503* varies between 30% and 93% depending on the ethnic population. The *C* allele is the minor allele in Asians, while *T* is the minor allele in Africans [306].

The results of studies investigating the effect of *rs2032582* are also questionable. Some studies support an association of the SNP with altered P-gp activity and expression, while others are opposed [307,308]. This allele is linked to an increase, decrease, and no change in drug exposure and drug effect [303,305,309]. The results of studies on disease risk are also conflicting. Research on inflammatory bowel disease, Crohn's disease, and ulcerative colitis has reported no genotypic effect of *rs2032582* [310,311]. Recently, however, a statistically significant association was found for *rs2032582* and steroid-resistant nephrotic syndrome [312]. The *rs2032582* allele frequency varies between 2–65% in world populations [300]. The frequency of the *2677 GG* genotype is 81% in African populations, while in American-Indians, Mexicans, Asians, and Caucasians, it is 10–32% [300].

Genetic variants in the ABCB1 gene *rs1045642* were associated with altered drug response and disease risk. However, the results of the investigations are controversial. While some studies have associated the *3435T* allele or *TT* genotype with decreased P-gp expression and increased drug levels, others have linked this genotype to the increased expression of P-gp or no genotypic effect at all [204,308,313–317]. Similarly, the *3435 CC* genotype was associated with increased drug concentrations or no genetic effect on the plasma drug concentrations [300,318–324]. In addition, a recent study did not find a significant association between SNP *C435T* and the pathogenesis of colorectal cancer. However, another study revealed a significant correlation of *rs1045642* with steroid-resistant nephrotic syndrome [312,325]. The allele frequency varies between different populations. The *3435C>T* allele frequency in the African population was estimated to be 83–84% for the C allele. The Caucasian, Southwest Asian, Chinese, and Saudi populations had lower frequencies of the C allele (48%, 34%, 53%, and 55%, respectively) [326].

The polymorphisms of the *ABCB1* gene were studied for their role in cannabis dependence [327,328]. The common SNP of *ABCB1* (*rs1045642)* was correlated with cannabis addiction [321,329]. Caucasian patients with cannabis dependence exhibited significantly higher *3435C* allele frequency and *CC* genotype compared to healthy controls [329]. It was suggested that *rs1045642* polymorphisms may affect THC distribution, psychoactive effect, and individual susceptibility to dependence [329], and the *CC* carriers may have an increased predisposition to cannabis addiction, while the *TT* genotype may have a greater

risk of cannabis-induced psychosis [329]. In another study, the *C3435T* polymorphism was studied in heavy cannabis users [330]. It was estimated that the *ABCB1 C3435T* polymorphism modulates THC blood levels and the *T* carriers (*TT/CT*) had significantly lower plasma THC concentrations than *non-T* carriers with the same weekly use. However, the exact mechanisms of the impact were not estimated [330].

**ABCG2.** The ABCG2 (BCRP) protein is a member of the ATP-binding cassette (ABC) transporter superfamily. Substrates of ABCG2 include topoisomerase inhibitors, anthracyclines, camptothecin analogs, tyrosine kinase inhibitors (TKI), antimetabolites, Aβ peptides, conjugates of steroids and xenobiotics, photosensitizers, and other compounds [331–334]. THC is also a substrate for the BCRP efflux transporter [335]. THC concentrations were higher in both Abcb1 (−/−) and Abcg2 (−/−) mice than in the wild-type. The knockout animals had prolonged elimination of THC from the brain, which was more noticeable in the Abcg2 (−/−) mice [335]. Moreover, the knockout mice were more sensitive to THC-induced hypothermia compared to the control mice [335].

*ABCG2* polymorphisms are known to contribute to multidrug resistance in cancer chemotherapy and have a correlation with survival rates and therapy response in cancer [333,336]. Previous studies have reported that variations in the *ABCG2* gene were associated with hyperuricemia, the prevalence and onset of gout, inflammation and autophagy, and some other disease states [337–344].

The effect of the *ABCG2* gene polymorphisms on the pharmacokinetics of multiple drugs has been demonstrated [345,346]. Upregulated *ABCG2* expression leads to a reduction in the drug plasma concentrations [347–349], while the downregulation and/or reduced-of-function variations tend to produce higher drug levels [345,350,351]. The majority of *ABCG2* polymorphisms are associated with a reduction in the overall *ABCG2* protein expression, and therefore reduced activity [352–354]. A common loss-of-function ABCG2 variant is *rs2231142* [355]. *rs2231142* has been associated with high uric acid/urate concentrations and gout development [344,356–358]. The *T* carriers of *Q141K* were linked to high risk of gout and reduced response to gout treatment by allopurinol [359,360]. However, recent studies have found no association between *rs2231142* and oxypurinol, or allopurinol riboside plasma concentrations [346,361]. In other studies, the *T* carriers not only produced high concentrations of other *BCRP* substrates such as rosuvastatin and imatinib, but also generated a greater therapeutic effect of the drugs [345,346,362–365]. Moreover, the SNP *C421A* may influence the susceptibility to cancer development, survival, and treatment outcomes [366–369]. A study showed a statistically significant correlation between the SNP *C421A* and the risk of multiple myeloma [368]. *rs2231142* produced worse outcomes in prostate cancer [370]. Prostate cancer patients with the *Q141K* variant had a shorter survival time than the wild-type carriers [370]. However, in other studies, *rs2231142* reduced the efflux of docetaxel in prostate tumors, resulting in improved drug response [370,371].

Additionally, an association between *ABCG2, 421C>A* and the development of Parkinson's and Alzheimer's diseases has been reported [367,372]. *ABCG2* was upregulated in the brains of Alzheimer's patients, and the *421CC* genotypes demonstrated a significantly increased predisposition to Alzheimer's disease compared to the *CA* and *AA* alleles [332,367,371]. Moreover, recent studies have reported that the *ABCG2* gene influences the susceptibility to psoriasis and blood glucose level in type 2 diabetic carriers. The heterozygote *GT rs2231142* individuals were less susceptible to psoriasis [356] and significantly higher glucose levels were in the type 2 diabetes patients with the *Q141K* variant [373]. The *rs2231142* polymorphism has a highly variable frequency depending on ethnicity. It is found commonly in Asian (26.6–35%) populations, but less frequently in Caucasian (8.7–14%), and African-American (up to 5.3%) populations [374,375].

*V12M rs2231137* is another frequent reduced function polymorphism of ABCG2 with a highly variable occurrence. This polymorphism was found with the highest incidences in Mexican-Indians (90%), Pacific Islanders (64%), and South-Eastern Asians (45%), but more rarely in Caucasian (2–10.3%), African-American (8.3%), and Middle Eastern populations (5%) [376–378]. The results of the *rs2231137* genomic studies are controversial.

Several studies have found no significant effect of *V12M* on urate transport and gout development [338,358], while other studies have reported that *V12M* had a protective impact against gout [379,380]. In cancer chemotherapy, the overall survival and clinical outcomes were improved in the *34AA/AG* genotypes in non-small-cell lung cancer, chronic myeloid leukemia, and renal cell metastatic cancer treated with tyrosine kinase inhibitors [381–385]. In contrast, the *34G>A* allele was associated with lower survival rates in pediatric acute lymphoblastic leukemia patients and diffuse large B-cell lymphoma [371,386,387]. Recently, the *rs2231137* polymorphism was also associated with a higher chance of drug-resistant epilepsy in children [388].

The *Q126X rs72552713* polymorphism is a rare loss-of-function polymorphism with no protein expression [377]. The *Q126X* polymorphism is missing in Caucasians and African-Americans [377]. Many studies have found a strong connection between the *Q126X* polymorphism and increased risk of developing gout [379,389]. A recent study demonstrated that a combination of the variations *Q126X rs72552713* and *Q141K rs2231142* were responsible for high concentrations of uric acid and increased the all-cause mortality in hemodialysis patients [344]. The *Q126X* polymorphism was also responsible for altered pharmacokinetics of other drugs [371].

Interestingly, a recent study demonstrated that neither THC nor 11-OH-THC was found to be a substrate or inhibitor of P-gp or BCRP at pharmacologically relevant concentrations. THC-COOH is a weak substrate and inhibitor of BCRP, but not of P-gp. It was concluded that P-gp and BCRP will not modulate the disposition of these cannabinoids in humans [390]. This result is very intriguing and requires further investigation.

Data on the effects of active transport polymorphisms on drug concentrations and therapeutic outcomes are controversial. The inconsistencies can be explained by the different localization of corresponding proteins in the cell membranes (basolateral versus luminal), which may affect drug concentrations in the blood and target tissues. Other factors are involvement of other transporters and, in some cases, the impact of metabolism on the drug concentration. The effect of the polymorphism of active transport is substrate specific and should be investigated on a drug-to-drug basis.

The successful use of pharmacogenomic testing with metabolizing enzymes and transporters is highlighted later in this review in the application section focused on epilepsy.

## 5. Other Genes of Interest

The National Institute on Health Abuse suggests that polymorphisms of catechol-O-methyltransferase (*COMT*) and alpha serine/threonine-protein kinase (*AKT1*) genes may affect the response to cannabis and predict the possible risk of psychosis and cognitive impairment [391,392].

**COMT.** *COMT* is a dopamine-metabolizing enzyme in the prefrontal cortex of the brain. Polymorphisms of the *COMT* gene have been associated with the risk of various neuropsychiatric diseases such as schizophrenia, panic disorder, bipolar disorder, and anorexia nervosa [393–397]. The most studied and common SNP in this gene is *Val158Met*, *rs4680*. The *Val158Met* significantly affects the expression and activity of the COMT enzyme. Val is a leading factor for high *COMT* activity, low synaptic dopamine levels, and altered prefrontal function [398]. Individuals with the *Val/Val* genotype have higher COMT activity and lower dopamine levels than carriers with other genotypes [399]. The *Met* variant corresponds to low enzymatic activity [400]. Allele frequencies of *Val108/158Met* polymorphism have been observed in three populations: Caucasians (0.28 *Met/Met*, 0.51 *Met/Val*, 0.21 *Val/Val* alleles), Asians (0.08 *Met/Met*, 0.42 *Met/Val*, 0.50 *Val/Val* alleles), and Africans (0.11 *Met/Met*, 0.41 *Met/Val*, 0.42 *Val/Val*) [401].

Genetic variants of COMT have been associated with the risk of cognitive impairment in cannabis users. A study demonstrated that the effect of THC on cognition and psychosis are moderated by the *COMT Val158Met* genotype. Carriers with the high-activity genotype *GG (Val/Val)* were more sensitive to THC-induced memory and attention impairments compared to carriers with the *Met* allele [400]. Another study linked rs4680 polymor-

phisms, cannabis use, and executive performance. Cannabis users carrying the *COMT Val/Val* genotype exhibited decreased attention, associated with a stricter response bias, and also committed more monitoring/shifting errors than cannabis users carrying the *AA (Met/Met)* genotype [402]. Moreover, *Val* allele carriers, but not subjects with the *Met/Met* genotype, more often showed more severe psychotic and schizophrenic symptoms and an increase in hallucinations after cannabis exposure [402–404]. In line with these studies, an investigation revealed that *COMT Val158Met* impacts the development of psychosis in people with at risk mental state (ARMS), particularly in weekly cannabis users [399]. This effect was increased in carriers with the *Val* allele and even more in *Val* homozygous individuals [399]. Additional studies have demonstrated the influence of the genotypes on cognitive functions upon THC administration. THC impaired working memory and attention in *COMT Val/Val*, but not *Met* carriers [400]. It also showed a significant interaction between *COMT* polymorphism and cannabis use on verbal fluency and speed of processing. The *Met* carriers had significantly better performance on both tasks compared to *Val/Val* homozygous [405]. The findings suggest that *Val* alleles were more sensitive to THC-induced cognitive, memory, and attention impairments and that the *COMT Val158Met* polymorphisms control the effect of cannabis use on the development and severity of subclinical psychotic symptoms.

*COMT* genetic variants have also been proposed to increase the risk of cannabis use disorders [406]. Interestingly, a case study demonstrated that schizophrenic subjects homozygous for the *Met* allele at *rs4680* had twice the increased probability of lifetime prevalence of cannabis use than *Val* homozygous carriers [407]. However, other studies did not confirm that the psychotomimetic and subjective effects of THC were influenced by the *COMT* genotype [194,408–411]. This divergence can be explained by the presence of gene–gene interactions as susceptibility to psychosis is mediated by several genes. Other reasons can be cannabis strains with different concentrations of THC and CBD, environmental factors related to psychotic risk, and study design. One study included only schizophrenic patients, while other investigations had only 1–2.6% patients with schizophrenia or schizophreniform disorder. This indicates the COMT–cannabis interaction may differ between schizophrenic patients and the general population [408–411]. Future studies are necessary, but currently, the evidence for the interaction remains unconvincing.

**AKT1**. *AKT1* is a gene encoding protein kinase, which is required for multiple cellular functions including dopamine signaling [392]. Polymorphisms in AKT1 (*rs1130233* and *rs2494732*) were associated with low brain *AKT* protein expression and the development of schizophrenia [412]. The level of protein AKT1 was 68% lower in patients with schizophrenia than in the controls [412]. A study reported that carriers with the *rs2494732 CC* genotype had decreased *AKT1* function and higher striatal dopamine release. These individuals demonstrated a greater than 2-fold increased chance of psychosis compared with carriers with the *TT* genotype [413,414]. A significant correlation was reported between the *rs2494732* genotypes and the frequency of cannabis use [391,413,415]. Moreover, *AKT1* was nominated as a marker of the genetic predisposition to psychosis in cannabis users [194,392]. Genetic variations in *AKT1* facilitate short-term as well as longer-term psychosis effects associated with the use of cannabis [414]. It was reported that *AKT1 rs2494732* mediates the acute response and dependence to cannabis and predicts psychotic reactions and schizophrenic symptoms in cannabis users [391,415]. Daily users with the *CC* genotype demonstrated a 7-fold increase in the odds of psychosis compared with the *TT* carriers [391,413]. Moreover, the *AKT1 rs2494732* genotype affects sustained attention reaction and accuracy measured by the continuous performance test (CPT) [414]. Cannabis users with the *CC* genotype were slower and less accurate in the CPT compared to *TT* carriers. Interestingly, cannabis users with the *TT* genotype had similar or better performance than non-using patients with a psychotic disorder [414]. A recent study provided additional evidence that *AKT1* modulates cognitive performance [416]. Analysis of the *AKT1* genotypes revealed that 35% of individuals were identified as an intermediate risk with the *C/T* genotype and 25% of patients were identified as high risk with the *C/C* geno-

type [194]. The following differences in the *rs2494732* allele frequency between populations were reported: Black Africans 0.42, Caucasians 0.46, and Asians 0.62 [413].

Genetic variations at AKT1 rs1130233 were found to regulate the functional brain activation and the short-term psychotic effects of cannabis [414,417]. Recently, it has been reported that the polymorphisms influence the neurofunctional effects of THC [418]. THC caused an increase in anxiety, transient psychotomimetic symptoms, and brain activation [412]. The significant increase in the brain activation by THC was associated with the variations in *rs1130233*, reduced *AKT1* gene expression, and altered methylation [412]. The number of *A* alleles at *AKT1 rs1130233* was linked to the THC effect on brain activation. The higher the number of *A* alleles, the greater the effect of THC on fear-related brain activation across a network of brain regions [418]. Another study reported a significantly reduced striatal activation and higher levels of psychotic symptoms produced by THC in *rs1130233 G* and *GG* carriers [417]. However, one study reported that *AKT1* does not modulate specific psychotomimetic response to cannabis [419]. The authors explained the inconsistency by the study design. The main difference was the measure of cannabis-induced psychotic-like experiences (cPLE). The late study used the modified cannabis experiences questionnaire (CEQ). Other studies used Psychotomimetic States Inventory PSI or did not measure the cPLE at all [419].

**Genome-wide association studies (GWAS).** The GWAS extended the list of related genes. A GWAS detected two genome-wide significant polymorphisms: FOXP2, *rs7783012* and EPHX2, rs4732724. The study reported that cannabis use disorder and cannabis use were genetically related. However, it was recommended partially by distinct genetic foundations of cannabis use and cannabis use disorder. Cannabis use disorder was also correlated with ADHD, major depression, and schizophrenia [420]. Another GWAS has identified eight significant independent SNPs. While no individual SNP achieved genome-wide significance, four genes were associated with lifetime cannabis use: NCAM1, CADM2, SCOC, and KCNT2. The greatest association with cannabis use had CADM2 (*rs2875907*, *rs1448602*, and *rs7651996*). This study also revealed an impact of schizophrenia on cannabis addiction and significant genetic overlap between cannabis and other substance use [421]. A meta-analysis of six GWAS revealed a new significant locus, *rs1409568* on chromosome 10, which was associated with the susceptibility to cannabis addiction. This study reported a modest support for the replication for *rs1409568* in African-Americans but not European-Americans. The combined meta-analysis suggested a trend-level significance for *rs1409568*. It was concluded that the discovery of this locus should be considered as preliminary [422].

Many additional genes have been associated with cannabis use disorders and cannabis induced changes in executive functions. The following genes have demonstrated a positive association: *DAT1*, *SLC6A4*, *DRD2*, *DRD4*, *BDNF*, *CHRM3*, *P2RX7*, *FAAH*, *ANKFN1*, *SLC35G1*, *CSMD1*, *ANKK1*, *COX2*, *ABHD6*, *ABHD12*, *MAPK14*, *SDK1*, *ZNF704*, *NCAM1*, *RABEP2*, *ATP2A1*, *ATP2C2*, and *SMG6* [46,402,406,417,423–435]. However, data on the effect of the polymorphisms of these genes are limited and controversial [419,429,436,437].

## 6. Cannabis Pharmacogenomic Applications and Personalized Medicine

Recent studies have started to elucidate the potential benefit for using pharmacogenomic testing to ascertain which individuals will derive positive effects from cannabis use and which individuals will encounter adverse events. Thus far, studies have been reported for pain management, epilepsy, and cannabis distribution, and consultation in community pharmacies.

### 6.1. Pain Management

With the increase in the use of cannabis in recent times, several positive attributes associated with its use have been identified. However, correspondingly, adverse effects have also been observed with some individuals. Interestingly, inter-individual variability has been observed with cannabis users and suggests that pharmacogenomic testing may help predict response. To assess the potential for pharmacogenomics to inform cannabis

pharmacotherapy, a study by Poli et al. (2022) focused on the use of cannabis in a population of chronic pain patients [438]. A total of 600 Italian patients were recruited to participate in an open label, multi-center non-randomized observational study to assess the association between cannabis treatment and chronic pain treatment. Participating patients were segmented into five groups based on their disease state: (1) central nervous disease; (2) arthritis and autoimmune diseases; (3) headache and migraine; (4) neuropathic; and (5) cancer. Six selected SNPs were selected for testing based on a TaqMan assay. The study demonstrated a 20% reduction in pain during the first month, with an overall decrease in pain to 43% after one year. However, a significant number of participants dropped out of the study due to poor or no pain reduction and/or side effects. There was a significant association between dropout and the polymorphism of the gene *CNR1*. The Poli study is the first reported study to demonstrate that certain polymorphic genes may be associated with a cannabis effect, both in terms of pain management as well as side effects.

### 6.2. Epilepsy

Although several treatment options are available for epilepsy, some epilepsies are associated with seizures that are resistant to existing treatment methods. Pharmacotherapy for pediatric epilepsy is particularly challenging; more effective therapies are needed to avoid short-term and long-term neurological disorders. Cannabis has been used to treat disease dating back to ancient times. Cannabis components, CBD and THC, are potential therapeutic options in epilepsy treatment. CBD has been shown to have an anticonvulsant effect in clinical studies. THC is the major psychoactive component of cannabis that contributes to the reduction in epileptic seizures. Concerns regarding the use of cannabis include the lack of standardization and regulation, imprecise dosing, possible adverse side effects, and drug interactions [439].

In the United States, approximately 3.5 million people have epilepsy [440], of these, twenty-five percent of the patients have treatment resistant epilepsy (TRE) [441]. Clearly, effective therapeutics are needed [442]. The use of pharmacogenomics should be able to identify predictors of CBD response. In a recent study, an open-label CBD study for TRE was executed using the Affymetrix Drug Metabolizing Enzymes and Transporters plus array [443]. A total of 113 patients participated in the study. The study demonstrated that genetic variation in pharmacogenes is associated with CBD response as well as the onset of adverse events in TRE.

### 6.3. Cannabis Use in a Community Pharmacy

In order to assess the potential of pharmacogenomic testing informing on the safe use of cannabis in the community pharmacy, a pilot study was performed at two urban pharmacies in Canada [194]. Twenty patients were pharmacogenomically profiled. Consultation was provided by pharmacists to the participants subsequent to testing. A total of 75% of the patients reported a high value in the pharmacist consultation. Additional studies will likely improve patient safety and allow individuals to make informed decisions regarding the use of cannabis.

## 7. Conclusions

This is a pivotal time for the integration of cannabis compounds into pharmacotherapy. Differences in research study outcomes reported in the literature have fueled the debate with regard to the potential benefits or harms that can be ascribed to the use of cannabis or its derivatives. In this report, we comprehensively studied cannabis compounds and the mapping of biomarkers that have been reported to date. The potential for cannabis compounds to be used in pharmacotherapy will be largely dependent upon the quality of the pharmacogenomic data. Some individuals/populations will benefit from cannabis compounds; others will not. The achievement of the complete human genome sequence in 2022 will enable more extensive pharmacogenomic studies to be performed [444]. The focus of near-term research needs to address: (1) key gaps in the evidence base with

attention to the pharmacogenomic, pharmacokinetic, and pharmacodynamic properties of cannabis; (2) establishment of standards to guide the generation of high-quality research; (3) development of conclusive evidence on the short- and long-term effects of cannabis compounds; (4) rigorously assess modes of delivery and dose–response relationships. The time has arrived for substantial research to be performed to provide comprehensive and conclusive evidence on the therapeutic effects of cannabis and cannabinoids.

**Author Contributions:** Both authors contributed equally to the manuscript. All authors have read and agreed to the published version of the manuscript.

**Funding:** This research received no external funding.

**Data Availability Statement:** The data presented in this study are openly available.

**Acknowledgments:** We would like to thank all of the peer reviewers and editors for their opinions and suggestions.

**Conflicts of Interest:** The authors declare no conflict of interest.

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
