# Peer review of "Cannabis Pharmacogenomics: A Path to Personalized Medicine"

_cimb, doi:10.3390/cimb45040228_

Round 1

Reviewer 1 Report

Comments and Suggestions for Authors

Babayeva & Loewy present an extensive review of genetic variants with pharmacogenomic profiles related to cannabis. I have suggestions I would like to address the authors:

1. On page 4, the authors mention "SNP at the rs806368 locus". rs806368 is not the locus. It is the SNP ID according to the dbSNP database.

2. I suggest the authors include the genomic position of the variants cited, using the hg38 reference genome and the transcript version used (i.e. NM_016083.6).

3. Relevant information I missed in the manuscript is the SNV consequence: 5'UTR, synonymous variants, missense, etc. I suggest including.

4. Variants' nomenclature should follow the guidelines of the HGVS, such as NM_016083.6:c.*3475=. 

5. I suggest the authors cite the PharmGKB database and the variants registered regarding cannabis pharmacogenomics.

6. Tables and figures would improve the manuscript's presentation. Some of the variants cited just once could be presented only in a table to help readers that are not from the genetics field to distinguish the most important variants in each gene.

7. Minor point: italicize all the genes mentioned in the manuscript.

8. Minor point: single nucleotide polymorphism (SNP) nomenclature has been changed to single nucleotide variant (SNV).

Reviewer 2 Report

Comments and Suggestions for Authors

It is an interesting paper trying to summarize the main findigns existing about pharmacogenomics of cannabis and its potential relationship to tailored clinical interventions. 

I have some comments

General: the manuscrit needs more structuration and synthetization of the information according the usefulneess in clinical context. Perhaps, it could be better for a cllinical Reader that the findgins reviewed will be associated with the pathologies: addiction, anti-inflamatory, ... or other clinical use. Firts, a brief introduction about mechanism of action, metabolism and genòmics should be incorporated. 

Introduction

I think that the rational of this part could be improved if first explain What is cànnabis as a pharmacological agent, secondly the clinical uses approved and after the importance to know genòmic aspects for clinical use.

Structuration of the review

There is a narrative review and no method was described. For mi is OK but, findings should be clearly and operative exposed (see my general comment, the authors only  make this in the final part, and I think that thew Reader only observe the usefulness of pharmacogenomics in epilepsy and pain managment). And, if it was the case, part of the information reviewed it is innecessary.

It should be mentioned as limitation of the paper that some of the papers reviewed need to be confirmed in posterior studies, due to limitations of the works (design, number of participants,....)

Reviewer 3 Report

Comments and Suggestions for Authors

The review Babayeva and Loewy on Cannabis pharamacogenomics is an interesting topic that has not been extensively reviewed, however there are several major issues with the article in its current format. The most extensive is the difficulty to read the article, it is very dense and there is no good cohesion to the article, it reads as a bullet list of identified SNPs in various metabolizing enzymes but doesn’t have any flow. 

Major Concerns

1.       The article references CBD and Epidiolex, CBDV and THC, but does not look at any of the FDA approved THC related medications (dronabinol and nabilone).

2.       The introduction is not well focused, and does a poor job setting up the review

3.       The involvement of COMT and AKT1 doesn’t really fit with the overall theme of the review.

4.       There is a lack of correlation between SNPs with functional binding or metabolism of the cannabinoids.  There seems to have been a focus on a large reference list, with citing a vast number of articles that do not relate to cannabis/cannabinoids, nor do the authors try to make any connections.

5.       Overall there is a lack of synthesis to the data reviewed. 

Minor Concerns

1.       The third paragraph in the TRPV section starts off with the impression it will be discussing TRPV1 variants, however there is mention of TRPV3 in the middle that is a bit out of place.

2.       The introduction states that THC is also metabolized by CYP2C19, however this is not mentioned in the first paragraph of the pharmacogenetics section

3.       The final paragraph of the 2C9 section would be better suited to the 2C19 section. 

4.       Use of inappropriate terminology, psychotic, Mexican Indians

5.       There is inconsistent use of underlining, highlighting, bold text throughout along with changes in font

6.       Line numbering would make reviewing and commenting much easier

Round 2

Reviewer 3 Report

Comments and Suggestions for Authors

Overall my concerns have been addressed